# Observational Study on the Clinical Reality of Community-Acquired Respiratory Virus Infections in Adults and Older Individuals

**DOI:** 10.3390/pathogens13110983

**Published:** 2024-11-09

**Authors:** Masayuki Nagasawa, Tomohiro Udagawa, Tomoyuki Kato, Ippei Tanaka, Ren Yamamoto, Hayato Sakaguchi, Yoshiyuki Sekikawa

**Affiliations:** 1Department of Infection Control, Musashino Red Cross Hospital, 1-26-1, Kyonan-cho, Musashino, Tokyo 180-8610, Japan; ict-ph2@musashino.jrc.or.jp (T.K.); i.tanaka@musashino.jrc.or.jp (I.T.); h.sakaguti@musashino.jrc.or.jp (H.S.); yoshiyuki1021@gmail.com (Y.S.); 2Department of Pediatrics, Musashino Red Cross Hospital, 1-26-1, Kyonan-cho, Musashino, Tokyo 180-8610, Japan; uda0112@yahoo.co.jp; 3Department of Pharmacy, Musashino Red Cross Hospital, 1-26-1, Kyonan-cho, Musashino, Tokyo 180-8610, Japan; 4Department of Laboratory, Musashino Red Cross Hospital, 1-26-1, Kyonan-cho, Musashino, Tokyo 180-8610, Japan; ron.0503.y@gmail.com; 5Department of Infectious Diseases, Musashino Red Cross Hospital, 1-26-1, Kyonan-cho, Musashino, Tokyo 180-8610, Japan

**Keywords:** respiratory virus, older adults, comprehensive PCR, comorbidity

## Abstract

The impact of common respiratory virus infections on adults and older individuals in the community is unclear, excluding seasonal influenza viruses. We examined FilmArray® tests performed on 1828 children aged <10 years and 10,803 adults, including cases with few respiratory symptoms, between January 2021 and June 2024. Approximately 80% of the children tested positive for ≥1 viruses, while 9.5% of the adults tested positive mostly for severe acute respiratory syndrome corona virus-2 (SARS-CoV-2). Besides SARS-CoV-2 infection, 66 out of 97 patients (68.0%) aged >60 years with rhinovirus/enterovirus (RV/EV), respiratory syncytial virus (RSV), parainfluenza virus-3 (PIV-3), or human metapneumovirus (hMPV) infection required hospitalization, of whom seven died; 26 out of 160 patients (16.3%) aged <60 years required hospitalization mostly because of deterioration of bronchial asthma, with no reported deaths. In older patients with RV/EV infection, three with few respiratory symptoms died due to worsened heart failure. Although the frequency of common respiratory virus infections in older adults is low, it may be overlooked because of subclinical respiratory symptoms, and its clinical significance in worsening comorbidities in older adults should not be underestimated.

## 1. Introduction

Respiratory virus infections are the most common infectious diseases and have the greatest impact on human hygiene and health [1,2,3,4,5]. The most obvious manifestation of this was the coronavirus infectious disease 2019 (COVID-19) pandemic, which not only caused many infectious disease-related deaths, especially in middle-aged and older adults, but also significantly affected socioeconomic activities worldwide [6,7,8,9]. Regarding respiratory virus infections, excluding influenza virus infections, children, especially infants, have been the main targets of research and observation. Immunologically immature children suffer more frequently from respiratory virus infections, and acquired immunity develops and matures consequent to repeated infections [10]. Furthermore, the structural characteristics of the respiratory and thoracic systems, such as narrow and short airways, elevated diaphragm, and horizontal rib structure [11,12], tend to make respiratory virus infections more severe in children, which is related to the fact that research on respiratory virus infections has been focused on children. 

Conversely, owing to the aging of society and increased number of patients with iatrogenic immunodeficiency due to advancements in medical technology, interest in and significance of respiratory virus infections in adults and older individuals are increasing [13,14,15]. Influenza virus infection is known to increase excess mortality in older adults during its epidemic season [16,17]. Although there are several reports that respiratory syncytial virus (RSV) infection is a risk factor for adults and older individuals in the intensive care unit (ICU) [18,19], there are few reports on the impact of common respiratory viruses on adults and older individuals in the community. COVID-19 posed a significant threat to adults and older individuals. Over the last ten years, comprehensive polymerase chain reaction (PCR) methods have been developed in research as diagnostic techniques for respiratory viruses, and their usefulness has been demonstrated in children [20,21,22,23]. The COVID-19 pandemic led to the rapid expansion of this technology in general clinical practice, providing an opportunity to examine respiratory virus infections in adults.

Between January 2021 and June 2024, our hospital performed comprehensive PCR testing: FilmArray® (FA; BioMerieux Japan, Tokyo) for respiratory viruses on 10,803 adults and 1828 children aged < 10 years as a clinical practice. We believe that by analyzing the results obtained, we could elucidate some aspects of general respiratory virus infections among adults and older individuals in the community.

## 2. Materials and Patients

### 2.1. Patients and Criteria for FilmArray® Test

All patients who underwent respiratory virus FilmArray® (FA; BioMerieux Japan, Tokyo) testing at our hospital between January 2021 and June 2024 were included. Respiratory virus FA test data were extracted from the clinical test database, and other necessary clinical information was extracted from electronic medical records. Regarding pediatric patients, tests were conducted on those who had fever or respiratory symptoms and were eligible for hospitalization. For patients with no obvious respiratory symptoms, only a severe acute respiratory syndrome corona virus-2 (SARS-CoV-2) quantitative antigen test was performed, and a respiratory virus FA test was not recommended, but the suitability of FA testing depended on the examining physician. For adults, respiratory FA testing was recommended for emergency inpatients until June 2023, regardless of the presence or absence of fever or respiratory symptoms, as SARS-CoV-2 infection prevention measures in the ward. Patients with fever and respiratory symptoms who visited the emergency department were provided a respiratory FA test depending on the clinician. Our hospital has been using the SARS-CoV-2 quantitative antigen test since December 2020, just like the FA test. However, during 2021, we prioritized the FA test because it was superior regarding diagnostic sensitivity. Due to the significantly increased number of individuals infected with the Omicron variant after the sixth wave in Japan, we decided to completely shift to quantitative antigen testing for COVID-19 diagnosis. From July 2023 onwards, quantitative antigen tests have been conducted for emergency patients, and respiratory FA tests were only recommended for inpatients with severe respiratory symptoms, depending on the examining physician.

### 2.2. Multiplex PCR Test

The FA (ver. 2.1) test was used to detect respiratory viruses in nasopharyngeal swab samples from patients, according to the manufacturer’s protocol. The panel test can detect 18 viruses: adenovirus (AdV); coronaviruses HKU1, 229E, OC43, and NL63; SARS-CoV-2; influenza A, A/H1, A/H1 2009, A/H3, and B; parainfluenza virus (PIV)-1, -2, -3, and -4; respiratory syncytial virus (RSV); rhinovirus/enterovirus (RV/EV); human metapneumovirus (hMPV); and four other microorganisms, including *Bordetella pertussis*, *Bordetella parapertussis, Chlamydia pneumonia*, and *Mycoplasma pneumoniae*.

### 2.3. Statistical Analysis

The statistical testing utilized Chi-square analysis, correlation analysis, Student’s *t*-test, and Mann–Whitney U analysis, and a *p* < 0.05 was considered significant. Statistical analyses were performed using JMP 14 software (SAS Institute, Cary, NC, USA).

## 3. Results

Table 1 shows the respiratory virus FA test results. Approximately 80% of children aged <5 years had ≥1 viruses detected, of which approximately 30% had multiple viruses. A maximum of six viruses were detected. These results were similar to those of previous studies [24,25,26,27]. Contrastingly, the overall positivity rate among adults aged ≥20 years was 9.5%. Among the positive cases, multiple viruses were detected only in 1.8%, with no cases in which ≥3 viruses were detected. Regarding the trends in the number of tests, as mentioned in the Methods section, many tests were conducted on adults between January 2021 and June 2022. No significant changes were observed in the distribution of age groups among adults tested during the study period (Figure 1).

Figure 2 shows the monthly number of tests and positivity rates for children (aged <10 years) and adults (aged ≥20 years). In children, although the number of tests fluctuated, the positivity rate remained almost constant. For adults, until June 2022, the number of tests was high, and positivity rate was low at 7.7%; however, from January 2023 onwards, the number of tests significantly decreased, and positivity rate increased to 29.1%. The dramatic changes in the number of tests and positivity rates among adults may have been due to changes in the indications for testing, as described in the Methods section.

Figure 3 shows the changes in the number and proportion of viruses detected. In adults, SARS-CoV-2 accounted for most positive cases in 2022, but significantly decreased with the transition from FA testing to quantitative antigen testing. In children, RV/EV was the most frequently detected virus, accounting for approximately 40% of the cases, followed by RSV, AdV, PIV-3, and hMPV (Table 2).

Regarding RV/EV, which has many positive cases, we investigated the correlation between the detection rates in children and adults (Figure 4). Between January 2021 and June 2022, as the number of tests for adults was extremely high, a significant correlation was observed (correlation coefficient, 0.66; *p* < 0.01). However, no correlation was observed after January 2023 (correlation coefficient, −0.02, *p* = 0.92). Moreover, this was true for RSV infections (correlation coefficient, 0.89 vs. 0.07). Frequent testing can detect unexpected respiratory virus infections.

To examine the clinical impact of respiratory virus infections in adults, we examined the clinical symptoms, course, comorbidities, and outcomes of individuals who tested positive for each virus. Comparative studies were conducted for adults, dividing them into age groups, 20–59 and ≥60 years, for each respiratory virus.

Table 3 shows the number of positive cases, hospitalized patients related to respiratory virus infections, and deaths from RV/EV, RSV, PIV-3, and hMPV infections among adults divided by age group. Patients who were hospitalized with a diagnosis apparently unrelated to respiratory virus infections, such as surgical diseases, epileptic seizures, and emergency deliveries, and who were positive for FA tests at admission were excluded from the admission list in Table 3. Among individuals infected with one of the four viruses, those aged ≥60 years were hospitalized at a significantly higher rate than those aged 20–59 years, and deaths were limited to those aged ≥60 years. Interestingly, even in individuals aged 20–59 years, patients with hMPV infection showed a high hospitalization rate. Seven patients aged ≥60 years died. Four of these patients died due to respiratory failure due to worsening of pneumonia, and three patients with RV/EV infection died due to worsening of heart failure. There were no cases of co-infection of RV/EV, RSV, PIV-3, or hMPV with other respiratory viruses in patients aged ≥60 years who required hospitalization. Among the patients who were urgently hospitalized with a diagnosis of diseases besides respiratory infectious diseases and no apparent respiratory symptoms, 35 were positive for RV/EV, four were positive for RSV in individuals aged 20–59 years, nine patients were positive for RV/EV, and one patient was positive for RSV and PIV-3 each, in those aged ≥60 years in the FA test upon admission.

We investigated the presence or absence of comorbidities in hospitalized and non-hospitalized patients aged ≥60 years infected with RV/EV and RSV (Table 4). Both groups showed a significantly higher incidence of comorbidities among hospitalized patients than non-hospitalized patients.

Figure 5 shows a consort diagram summarizing the effects of four common respiratory virus infections on adults and the older individuals revealed in this study.

## 4. Discussion

There has been little clinical interest in respiratory virus infections excluding seasonal influenza in adults. Influenza has been shown to directly and indirectly increase excess mortality in older adults, demonstrating the importance of vaccination [28,29,30]. Conversely, in the field of pediatric infectious diseases, respiratory virus infections are the biggest and most important clinical issue, and the development and clinical application of preventive treatments for RSV infection, which is among the most serious respiratory infections in infants, are progressing through monoclonal antibody preparations [31,32] and vaccine development [33,34]. Antigen test kits have been developed as point-of-care tests (POCT) for influenza, adenovirus, RSV, and hMPV, and are widely used in daily medical practice [35,36,37]. Since 2010, methods for comprehensively testing various respiratory viruses using PCR have been developed in clinical research, and their usefulness has been reported [20,21,22]. Contrastingly, the COVID-19 pandemic has reminded us of the importance of common respiratory virus infections not only in children but also in adults, and diagnostic methods, such as the abovementioned, are currently used for adults. Particularly, it has been used as an emergency evacuation tool in abnormal situations, such as pandemics, for adults and older individuals. However, as a by-product, the actual status of common respiratory virus infections in adults and older individuals in the community may be estimated, which was previously unclear.

Our observational study showed that respiratory virus infections with RV/EV and RSV, which are prevalent in children, are endemic in adults, albeit less frequently. Adults are more likely to have mild or lesser symptoms than children. Although this was assumed to some extent previously [38], we believe that the significance shown by observational research is important.

Our study has shown that RV/EV, which has not received much attention as a common cold virus, can cause serious illnesses, including death, in patients aged ≥60 years, similar to RSV, PIV-3, and hMPV infections. Furthermore, in three out of four deaths, the cause was worsened comorbidities, such as heart failure. Among the 26 hospitalized older patients with RV/EV infections and comorbidities, heart failure was the main diagnosis in seven patients. Older patients whose main diagnosis was heart failure were not observed among the hospitalized patients infected with RSV, PIV-3, or hMPV in our cohort. RV/EV is a non-seasonal respiratory virus infection in children unlike RSV, PIV-3, and hMPV infections [39,40,41]. Conversely, the explosive spread of seasonal influenza and its impact on excess deaths among older adults are relatively easy to understand, but the impact of RV/EV on excess deaths among older adults is believed to be less obvious epidemiologically because it is a non-seasonal infectious disease. The reason for no positive RV/EV cases being detected among older individuals hospitalized with worsening heart failure after 2023 may be that FA testing was not performed because they did not have any respiratory symptoms. As a fact that supports this idea, the number of pediatric RV/EV infections increased contradictorily compared to that during 2021–2022. Recently, acute cardiac events have been reported to frequently occur in elderly patients with RSV infections [42]. The extent to which RV/EV infection is involved in worsening heart failure in older adults should be further investigated. RV/EV, RSV, and PIV-3 infections in adults aged 20–59 years were relatively mild, and there were almost no cases of hospitalization owing to pneumonia. Although the number of cases was small, most hospitalizations due to RV/EV, RSV, and PIV-3 infections in those aged 20–59 years were due to deterioration of bronchial asthma. RV/EV was detected in >90% of hospitalized children with bronchial asthma attacks in our cohort, which shows some similarities to young adults. Contrastingly, hMPV infection resulted in a high rate of hospitalization, even in this age group.

This study has several limitations. First, it was a retrospective observational study. Second, the most serious issue was that the indications for FA testing depended on the attending physician and, hence, standards are unclear.

The advantage is that under the circumstances of the COVID-19 pandemic, FA testing has been used widely as a part of nosocomial infection control for adults, especially older adults, and for asymptomatic hospitalized patients for approximately 18 months (Figure 1 and Figure 2). By comparing the results from January 2023 onward, as the eligibility criteria for FA testing for adults were similar to those for children, the actual situation of community-acquired respiratory virus infections in adults and older individuals was examined in greater depth. Between January 2021 and June 2022, approximately 9000 FA tests were conducted on adults. Eventually, it will be extremely difficult to conduct a large number of tests on asymptomatic or less symptomatic patients as a part of a new clinical research at a single center in the future.

Although there are some issues with the research design and eligibility criteria for FA testing, we believe that the study observations present important findings and suggestions regarding the impact of common respiratory virus infections on the health of adults and older individuals.

## 5. Conclusions

The frequency of common respiratory virus infections in older adults is low compared to that in children. However, common respiratory viral infections in older adults may be overlooked because of subclinical respiratory symptoms, and their clinical significance in worsening comorbidities to deaths in them should not be underestimated.

## Figures and Tables

**Figure 1 pathogens-13-00983-f001:**
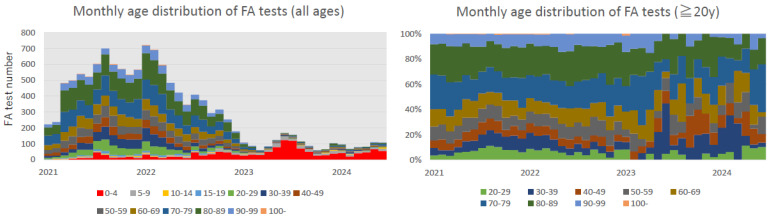
The monthly number of FilmArray® (FA) tests is shown by age group. The left figure shows the changes in the absolute number of tests. The figure on the right shows the trends in the relative proportion to the number of tests conducted on individuals aged ≥20 years.

**Figure 2 pathogens-13-00983-f002:**
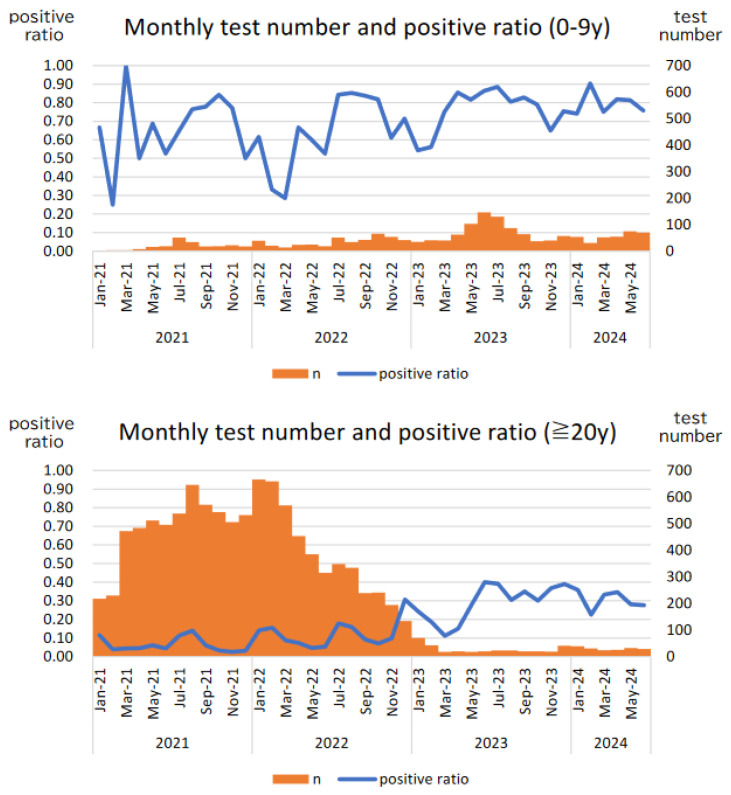
The figure on top shows the monthly trends in the number of FilmArray® (FA) tests for children (aged <10 years) and the positivity rates for cases in which ≥1 respiratory virus was detected. The figure on the bottom shows the monthly number of tests and positivity rates for individuals aged ≥20 years.

**Figure 3 pathogens-13-00983-f003:**
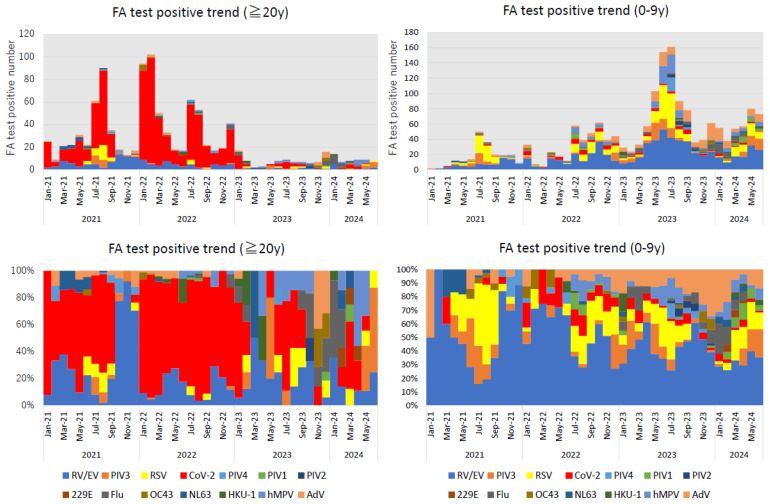
The absolute values (**top**) and relative proportions (**bottom**) of the number of respiratory viruses detected monthly. The figures on the left and right sides show the results for adults (aged ≥20 years) and children (aged <10 years), respectively.

**Figure 4 pathogens-13-00983-f004:**
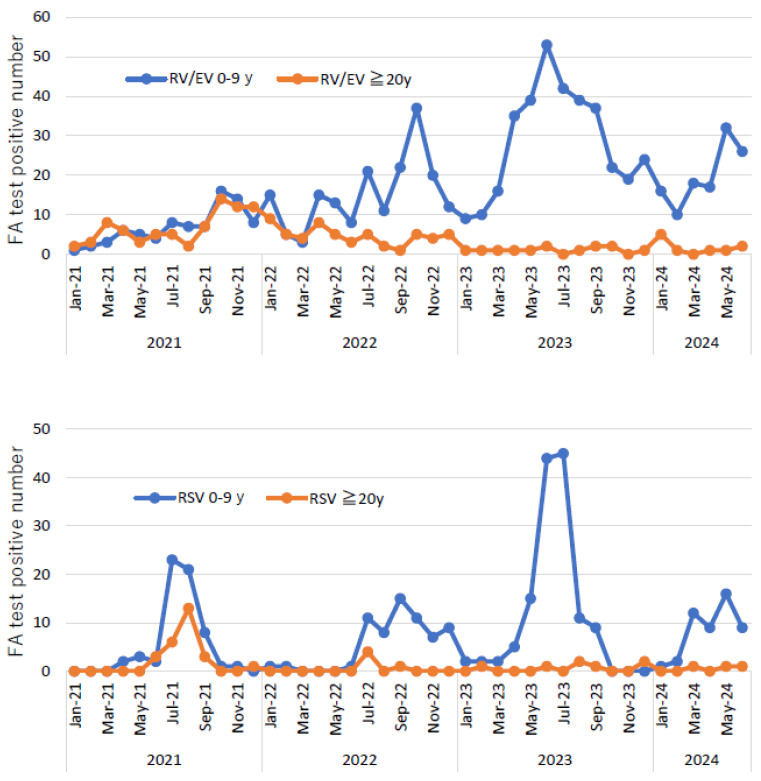
Monthly trends in rhinovirus/enterovirus (RV/EV) (**top**) and respiratory syncytial virus (RSV) (**bottom**) cases detected in children (aged <10 years) and adults (aged ≥20 years). There is a strong correlation between the trends in the number of virus cases detected in children and adults between January 2021 and June 2022, but no correlation is noted after January 2023.

**Figure 5 pathogens-13-00983-f005:**
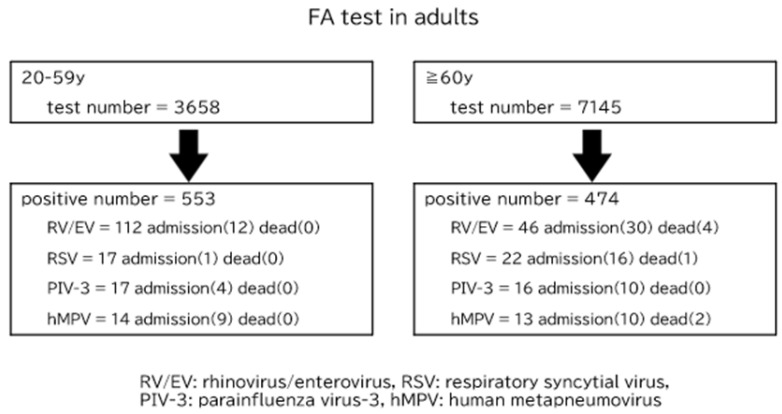
Consort diagram summarizing the effects of four common respiratory virus infections on adults and the older individuals in this study.

**Table 1 pathogens-13-00983-t001:** The number of FilmArray® tests by age, number of positives, positivity rate, detected cases of multiple virus, and their ratios.

Age (Years)	Test (n)	Positive (≥1)		Multipe Positive	
0–4	1479	1196	80.9%	357	29.8%
5–9	349	230	65.9%	45	19.6%
10–14	130	41	31.5%	4	9.8%
15–19	178	38	21.3%	4	10.5%
20–29	774	150	19.4%	5	3.3%
30–39	971	148	15.2%	2	1.4%
40–49	804	132	16.4%	1	0.8%
50–59	1109	123	11.1%	3	2.4%
60–69	1280	89	7.0%	2	2.2%
70–79	2218	150	6.8%	2	1.3%
80–89	2639	168	6.4%	3	1.8%
90–99	981	64	6.5%	0	0.0%
≥100	27	3	11.1%	0	0.0%

**Table 2 pathogens-13-00983-t002:** The number and frequency of respiratory viruses detected in children (aged <10 years) and adults (aged ≥20 years).

Respiratory Virus	0–9 Years		≥20 Years	
	(n)		(n)	
RV/EV	727	39.8%	158	1.5%
RSV	309	16.9%	39	0.4%
PIV3	157	8.6%	33	0.3%
hMPV	120	6.6%	27	0.2%
CoV-2	87	4.8%	677	6.3%
AdV	207	11.3%	28	0.3%
PIV1	29	1.6%	4	0.0%
PIV2	16	0.9%	1	0.0%
PIV4	33	1.8%	4	0.0%
Flu	53	2.9%	26	0.2%
229E	6	0.3%	3	0.0%
OC43	25	1.4%	12	0.1%
NL63	19	1.0%	17	0.2%
HKU-1	9	0.5%	11	0.1%
test number	1828		10,803	

RV/EV: rhinovirus/enterovirus, RSV: respiratory syncytial virus, PIV-1,2,3,4: parainfluenza virus-1,2,3,4, hMPV: human metapneumovirus, CoV-2: severe acute respiratory syndrome corona virus-2, AdV: adnovirus, Flu: influenza virus, 229E, OC43, NL63, HKU-1: coronavirus 229E, OC43, NL63, HKU-1.

**Table 3 pathogens-13-00983-t003:** Comparison of hospitalization rates for detected respiratory virus cases in adults aged 20–59 and ≥60 years.

Respiratory Virus	Age Group	n	Age (mean ± SD)	Admission (n)	Admission Rate	Death
RV/EV	20–59 years	112	36.9 ± 9.3 years	12	10.7%	0
	≥60 years	46	73.4 ± 14.6 years	30	65.2%	4
RSV	20–59 years	17	40.1 ± 11.0 years	1	5.9%	0
	≥60 years	22	78.3 ± 9.1 years	16	72.7%	1
PIV3	20–59 years	17	37.8 ± 10.3 years	4	23.5%	0
	≥60 years	16	76.6 ± 9.2 years	10	62.5%	0
hMPV	20–59 years	14	44.6 ± 7.4 years	9	64.3%	0
	≥60 years	13	75.0 ± 7.3 years	10	76.9%	2

RV/EV: rhinovirus/enterovirus, RSV: respiratory syncytial virus, PIV-3: parainfluenza virus-3, hMPV: human metapneumovirus.

**Table 4 pathogens-13-00983-t004:** Comparison of hospitalization rates based on the presence or absence of comorbidity in patients aged ≥60 years with respiratory viruses.

		n	Comorbidity	(%)	Comorbidity (%)				
					Chronic Respiratory Diseases	Heart Diseases	Neurological Diseases	Diabetes	Cancers
RV/EV	admission	26	Yes	86.70% *	57.7%	42.3%	10.0%	26.9%	11.5%
		4	No						
	non-admission	9	Yes	56.30%	33.3%	11.1%	12.5%	0.0%	33.3%
		7	No						
RSV	admission	15	Yes	93.8% **	53.3%	25.0%	46.7%	20.0%	20.0%
		1	No						
	non-admission	3	Yes	50.0%	100.0%	33.3%	33.3%	0.0%	0.0%
		3	No						

RV/EV: rhinovirus/enterovirus; RSV: respiratory syncytial virus; *: Odds ratio 5.06, 95% CI 1.19–21.4, *p* = 0.021; **: Odds ratio 15.0, 95% CI 1.14–198, *p* = 0.018.

## Data Availability

Data available on request from the authors. The raw data supporting the conclusions of this article will be made available by the authors on request.

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
