# Peer review of "Observational Study on the Clinical Reality of Community-Acquired Respiratory Virus Infections in Adults and Older Individuals"

_pathogens, 2024, doi:10.3390/pathogens13110983_

Round 1
Reviewer 1 Report
Comments and Suggestions for Authors
The article "Observational study on the clinical reality of community-acquired respiratory virus infections in adults and older individuals" explores a timely topic, making pertinent observations during a period when the impact of COVID-19 on the healthcare system was significant. The study is original, and the discussions are relevant. For improvement, I suggest:
- Reducing the similarity index
- Figure 3 is difficult to follow
- It is necessary to include a flow chart for proper visualization of the study
- Considering the large sample size and the subdivision into many age groups, a more detailed statistical analysis is required
- Clear mention of the inclusion and exclusion criteria for the study.
Author Response
Dear Reviewer 1
Please see the attachment.

Reviewer 2 Report
Comments and Suggestions for Authors
In the manuscript submitted to me for review entitled "Observational study on the clinical reality of community-acquired respiratory virus infections in adults and older individuals“ the authors Masayuki Nagasawa, Tomohiro Udagawa, Tomoyuki Kato, Ippei Tanaka, Ren Yamamoto, Hayato Sakaguchi and Yoshiyuki Sekikawa have conducted a study in the period January 2021 and June 2024 in which, using the FilmArray® test, they tracked the spread of respiratory viral infections among the population in different age groups.
The obtained results are presented in detail with the help of 4 figures and 4 tables. To support their research, the authors used 42 references that present information from studies published mostly in the last fifteen years. About 2/3 of the total references are from the last 5 years, indicating that the spread of viral respiratory infections is a topic that is attracting more and more attention from researchers especially after the COVID-19 pandemic. This means that the current manuscript definitely presents information that would be of interest to Pathogens readers. I did not notice any redundant self-citations, all the references used are appropriate and necessary for the preparation of the manuscript.
My remarks and recommendations to the authors are:
1. In Figures 1, 3 and 4, the ordinates of the subfigures do not indicate what the given values ​​actually represent.
2. Table 2 does not state that 229E and OC43 are coronavirus strains.
3. Table 2 presents an interesting fact that is not commented on anywhere in the text. Of the total number of children aged 0 - 9 years with a diagnosed coronavirus infection, 1/4 are with the old strains 229E and OC43. It would be good to comment on this fact somewhere.
Author Response
Dear Reviewer 2
Please see the attachment.

Reviewer 3 Report
Comments and Suggestions for Authors
Authors attempt to address important topic of impact of respiratory virus infections besides influenza. Comments as follows:
-The authors need to discuss the burden of disease that has been established prior to COVID era as that would help compare impact of COVID on other viruses.
-Recommend authors to discuss sensitivity, specificity, false positive and false negative rates using the test mentioned in the study that is established in literature.
-Authors need to discuss the exact methodology for exclusion of patients that were tested but suspected respiratory failure to be due to alternate etiology. What was the number of patients that were excluded due to this reason? How was the determination of exclusion conducted?
-Authors need to provide a consort diagram describing the study results starting from patients screened.
-Do authors have data on PCR based testing for patients in with acute respiratory failure who underwent alternate invading testing like bronchoscopy
-Authors need to discuss, in cases where multiple viruses were detected, how was the etiology determined?
-Authors need to compare their results with established literature prior to COVID era and how potentially COVID based mandates such as social distancing measures/masking may have affected their results
-Authors need to provide information about immunocompromised status of individuals included in the study since they are prone to severe infections.
Comments on the Quality of English Languagegood
Author Response
Dear Reviewer 3
Please see the attachment.

Round 2
Reviewer 1 Report
Comments and Suggestions for Authors
The article has been improved according to the requirements, but some figures and tables are outside the page margins and need adjustments.
Author Response
Dear Reviewer #2
We appreciate your kind 2nd review of our manuscript and valuable recommendations.
In accordance with the reviewer's remarks, Figures 1, 3, and 4 were replaced with new ones with explanations of the Y-axis.
The arrangement of the figures and tables has been left as it is because of the disorder of past proofreading and correction, so please forgive us if some of them are a little difficult to see.
We sincerely hope that this revision will be accepted.
Corresponding author:
Masayuki Nagasawa, M.D., Ph.D.,
Department of Pediatrics, leader of Infection Control and Antimicrobial Stewardship Team,
Musashino Red Cross Hospital
1-26-1, Kyonan-cho, Musashino-city, Tokyo 180-8610, Japan
Tel +81-0422-32-3111 Fax +81-0422-32-9551
e-mail : mnagasawa.ped@tmd.ac.jp masayukin@musashino.jrc.or.jp